# How do researchers approach societal impact?

**Benedikt Fecher** [1,2]* , **Marcel Hebing** [1,3]

**1** Alexander von Humboldt Institute for Internet and Society, Berlin, Germany, **2** German Institute for Economic Research (DIW Berlin), Berlin, Germany, **3** DBU Digital Business University of Applied Sciences, Berlin, Germany

These authors contributed equally to this work.
* fecher@hiig.de

**Data Availability Statement:** The data underlying this study are available on Kaggle (DOI: 10.34740/kaggle/dsv/2366092).

**Funding:** B.F. received funding by the German Federal Ministry of Education and Research (01PW18008A; 01PW18008B; https://www.

## Abstract

Based on a communication-centered approach, this article examines how researchers approach societal impact, that is, what they think about societal impact in research governance, what their societal goals are, and how they use communication formats. Hence, this study offers empirical evidence on a group that has received remarkably little attention in the scholarly discourse on the societal impact of research—academic researchers. Our analysis is based on an empirical survey among 499 researchers in Germany conducted from April to June 2020. We show that most researchers regard societal engagement as part of their job and are generally in favor of impact evaluation. However, few think that societal impact is a priority at their institution, and even fewer think that institutional communication departments reach relevant stakeholders in society. Moreover, we show that researchers' societal goals and use of communication formats differ greatly between their disciplines and the types of organization that they work at. Our results add to the ongoing metascientific discourse on the relationship between science and society and offer empirical support for the hypothesis that assessment needs to be sensitive to disciplinary and organizational context factors.

## Introduction

Until the 1970s there was no doubt among policymakers that public investment in research would have a positive societal impact. The social contract for science at that time meant that science was granted an unusual degree of autonomy in return for widely diffused benefits to society and the economy [1]. It was only from the late 1980s onward that researchers were increasingly expected to account for their achievements in evaluation exercises [2–4]. Initially, these focused on intra-scientific (and often bibliometric) impact. In the last decade, however, policymakers have begun to focus more on the societal impact of research and hence on what academic research offers for the economy, society, culture, public administration, health, environment, and overall quality of life [2, 5, 6]. Noteworthy examples indicating the gradual shift from scientific to societal impact in research governance include the Research Excellence Framework (REF) in the United Kingdom or the Excellence in Research for Australia

wihoforschung.de/de/impaqt-2631.php). The funder had no role in study design, data collection and analysis, decision to publish, or preparation of the manuscript.

**Competing interests:** The authors have declared that no competing interests exist.

Framework (ERA) [7–9]. In the Netherlands, a region with some of the most developed examples of impact governance, the Association of Universities in the Netherlands (VSNU), the Netherlands Organisation for Scientific Research (NWO), and the Royal Netherlands Academy of Arts and Sciences (KNAW) have implemented guidelines for the evaluation and improvement for research, the Standard Evaluation Protocol (SEP). The SEP is used by research institutions to evaluate research units and focuses strongly on relevance for society as well as research quality [10, 11]. Societal impact is also a key component in European research funding [12–14].

In Germany, where the present study was conducted, there is no evaluation exercise comparable to the REF, ERA, or SEP. The topic has, however, been prominently discussed: Referring to the announcement that the next EU Framework Programme for Research and Innovation will focus more on the societal benefits of research, the Alliance of Science Organisations stated in 2018 that any such evaluation should be differentiated and tailored to the demands of science [15]. In 2019, the Federal Ministry of Education and Research (BMBF) published a policy paper in which it stipulated that societal impact must become part of the academic reputation logic [16]. In 2020, it set up a think tank to work out how societal impact could be evaluated [17]. Also in 2020, the Alliance of Science Organizations issued an agreement containing four fields of action; this highlighted, among other things, scientific freedom and the need to anticipate disciplinary differences. The German Council of Sciences and Humanities, an advisory body to the German federal government, called for "more recognition for knowledge and technology transfer" [18]. The German Rectors' Conference, the umbrella organization of German universities, decided at its general assembly (November 14, 2017) that knowledge transfer is a priority for universities [19]. Besides these developments at the level of science governance in Germany, there are a multitude of institutional initiatives and strategy developments underway. For instance, the Leibniz Association, a union of 96 nonuniversity research institutes, adopted a new mission statement for the transfer of scientific knowledge to society, the economy, and politics in 2019 [20].

Although there is no evaluation exercise in Germany comparable to the REF in the UK, it is evident that the topic has gained momentum in Germany in recent years as well. In light of complex societal challenges and the further integration of German research bodies into the European research area, the societal impact of research will likely become an even more prominent concern in research governance in the near future. If, as stated above, the old social contract granted science relative freedom in return for widely diffused benefits for society, the new social contract for science imposes accountability for the freedom granted to science [21–23]. This development has been criticized by the scientific community. For example, many associations for the social sciences and humanities in Germany expressed concern that the BMBF initiative does not take into account the current state of science communication research [24].

If societal impact is to be a new paradigm of science governance, it is important to better understand how societal impact emerges. In this article, we address this question from the perspective of researchers, that is, we ask what their views on societal impact are, what their goals are, and what formats they use to achieve them. Our analysis is based on a survey among 499 researchers in Germany conducted from April to June 2020. Here, we report the key results of this study and reflect on potential implications for science governance. Our results add to the ongoing scholarly discourse on societal impact by offering empirical evidence on a group that has received surprisingly little attention in the scholarly study of societal impact—academic researchers.

## Societal impact in research

Two scholarly discourses are particularly relevant for the subject of this study: 1) the discourse in communication and science and technology studies (STS) on the relationship between science and society and 2) the discourse on societal impact measurement in scientometrics and evaluation research. In the following, we will delineate these two discourses and reflect on their implications for this study.

**Relationship between science and society: From deficit to dialog.** Before the first large-scale impact agendas were implemented, scholars in STS critically examined the nature and role of science in society, drawing on novel concepts of academic knowledge creation such as "Mode 2" [25, 26], "academic capitalism" [27], "post-normal science" [28] or "Triple Helix" [29]. Although these concepts differ in their objectives, they generally assume a form of scientific value creation that is no longer self-sufficient and is increasingly interwoven with society. While these conceptualizations do not employ societal impact as an evaluative paradigm, they have paved the way for new thinking about the relationship between science and society. These explorations of the role of science in society were, however, primarily theoretical in nature. Later, entire lines of (communication) research addressed how the public deals with science, for example, the public understanding or awareness of science (PUS, PAwS), scientific literacy, or more recently the public engagement with science and technology (PEST) and the science of science communication [30–33]. In general, research in this area has shifted away from viewing society as an inactive recipient of knowledge—for example, in the so-called "deficit model of science communication" [34]—and towards envisaging more complex and interactive forms of knowledge creation and dissemination [3, 35–37].

While there is a considerable body of literature on how the public perceives research, less attention has been paid to the institutional conditions for engagement and how researchers themselves deal with the public. Regarding these institutional conditions, scholars have observed a certain decoupling of central communication infrastructures at institutions and the researchers working there [38, 39]. Others have pointed to the increased legitimation pressure exerted by research organizations and the increase in PR and marketing [40]. When it comes to researchers' dealings with the public, scholars have focused on a) the relationship between science and specific publics—for instance, the media [41, 42] or politics [41], b) the relationship between science and the broader public [41, 43–45] or c) the communication practices of single disciplines [46, 47]. Here, recurring themes include researchers motivations for engaging with the public [48, 49], teaching and training [50, 51], and institutional conditions [52, 53]. Much research on the interfaces between science and society consists of various relevant case studies. Yet, there is a lack of comparable empirical evidence on one particularly decisive group of actors in the dialogical rationale, that is, academic researchers. This explains why, in this study, we focus on researchers and how they approach societal impact.

**Measuring societal impact: From economic to broader societal benefit.** In scientometrics and the wider field of evaluation research, the shift towards societal impact as an assessment paradigm in science governance has been accompanied by critical reflections. Martin, for example, asked pointedly whether the creation of the REF would create a "Frankenstein monster," because the costs of conducting the evaluation might outweigh the benefits [8]. Others have argued that evaluations of societal impact are prone to methodological shortcomings [23, 54] and might have unintended behavioral effects [55, 56]. In addition, there are two further points of criticism concerning the concept of societal impact: On the one hand, critics point to the inadequate representation of the complexity of science, for example, because impact logics of the natural sciences are used as a yardstick for evaluations [6, 57]. In contrast, impact assessments have been described as failing to do justice to transfer activities in the

social sciences and humanities (SSH) [58–60]. In recent years, new approaches have been developed that specifically address the SSH [6, 61, 62]. On the other hand, criticism is directed towards the representation of society, whose benefit has often been reduced to economic indicators (e.g., revenue, jobs) and not the broader societal impact [5, 63]. However, some evaluation exercises have broadened their scope to include the wider benefits to society [2].

Recent models for societal impact have tried to incorporate dialog into their conceptions of societal impact. An example of this is Jong et al.'s concept of productive interactions, which proceeds from the assumption that current productive interactions between researchers and societal stakeholders will improve the probability of future societal impact [64]. In this regard, interactions are deemed productive if encounters between researchers and societal stakeholders lead to knowledge that is academically sound and socially valuable [62, 64]. The concept indicates that two quality regimes—a scientific and a societal—are important for deciding on the productiveness of a dialog. It should be noted that science is part of society and both the sciences and their publics have differing ideas about robustness and usefulness [32]. The notion of societal impact as an (effect of) dialog is useful in the context of this study, because it allows us to move from pure utilization and reception research to examining the anticipated societal impact of researchers. Drawing from the wider field of evaluation research, we examine how academic researchers anticipate the broader societal impact of their research.

## Conceptualization of societal impact in this study

Building on the discourse in evaluation research, we refer to broader societal impact, i.e., the benefits that research holds for the economy, individual wellbeing, the environment, and culture [2, 14]. According to Bornmann, three main strands of societal impact can be distinguished: First, societal impact as a product (i.e., as an artifact that embodies scientific knowledge), second, societal impact as use (i.e., the adoption of academic knowledge by societal stakeholders), and third, societal impact as benefits (i.e., the effects of the use of research) [2]. Here we focus on the latter—we consider the desired societal benefits that researchers associate with their work (goals, RQ2) and the formats they use to communicate about their research (formats, RQ3). To further differentiate societal benefits and formats, we conducted a qualitative prestudy (see Method section). To gain a better understanding of how researchers perceive societal impact as a paradigm in research governance, we included questions to elicit respondents' opinions on the role of societal impact at their institutions and in their work, on whether societal impact should be given more weight in evaluations, and on whether their institutional communication departments are reaching relevant societal stakeholders [38,39] (opinions, RQ1).

## Explanatory dimensions

Drawing on De Jong and colleagues and in line with the metascientific discourse on the conception of the relationship between science and society, we see societal impact as the effect of interaction between scientific and societal stakeholders [61, 62, 64]. We hence used a framework for communication inspired by Cohn's concept of theme-centered interaction [65, 66] and Luhman's notion of meaning [67, 68] when deriving our explanatory variables. We differentiated between three dimensions of explanatory variables:

- The **content dimension** is defined by the researcher's disciplinary background and their self-assessment as to whether their research is applied or basic. It can be assumed that researchers' approaches to societal impact varies between disciplines [58–60]. We assumed that a researcher's disciplinary background would affect their choice of societal goals. We

further expected to find differences between researchers who considered themselves applied and those who considered themselves basic researchers—societal impact might play a greater role for applied researchers [69–73]. We therefore assumed that applied researchers would be more supportive of societal impact evaluations and would also be more active in using communication formats.

- The **organizational dimension** is defined by the type of research organization (i.e., universities, universities of applied sciences, nonuniversity research institutions). The organizational factors influencing societal impact have, with notable exceptions [52, 53], been little researched so far. Here, we were particularly interested in differences between various types of public research institutions in Germany (see Method section below). We assumed that researchers at independent institutes would be more active than university researchers in using communication formats because they usually do not have teaching obligations. We further expected researchers who worked at applied science universities to be more active in using collaboration formats.

- The **individual dimension** is defined by the sociodemographic factors of age, gender, and academic status. Gender and age differences in connection with human agency have been widely researched in the social sciences, also as they pertain to scholarly communication [74–80]. How these relate to researchers' engagement with society is still little understood. Academic status may have an influence on the use of communication formats, in the sense that high-status academics may be most active in advisory roles.

## Research questions

By looking at these dimensions, we aimed to understand how these factors might influence the interaction between scientific and nonscientific actors in terms of the opinions researchers hold, the societal goals they pursue, and the formats they use to communicate about their research. Three research questions guided our analysis and structured the presentation of the results; when analyzing each question, we use our three dimensions of explanatory variables (i.e., content, organization, individual) to structure and compare the results.

- **RQ1**: What are researchers' **opinions** on societal impact? What differences can be identified along the three dimensions (i.e., content, organization, individual)?

    In particular, we were interested in researchers' perspectives on a) societal engagement as a part of scientific activity, b) whether societal impact should be considered more in research evaluation, c) the performance of institutional communication departments in reaching relevant societal stakeholders, and d) the importance of knowledge transfer at the institutions.

- **RQ2**: Which **societal goals** do researchers aim to achieve with their research? How do these differ along the three dimensions (i.e., content, organization, individual)?

    We used the 13 goals that we identified by coding the REF use cases as a framework (e.g., supporting legislative decision making, driving technical innovation, preserving cultural heritage, or protecting the environment).

- **RQ3**: Which **formats** do researchers use to achieve societal impact? How do these differ along the three dimensions (i.e., content, organization, individual)?

    Again, we used our coding of the REF use cases as a framework and took a close look at educational offerings, consulting, events, PR, social media, and collaborations.

## Method

The institutional review board and data protection officer of the Alexander von Humboldt Institute for Internet and Society approved this study. Informed written consent was obtained from the participants in this study. The data was analyzed anonymously. To answer the research questions, we conducted a survey from April to June 2020 among 499 scientists in Germany. In the following, we describe the instrument and its distribution, the sample design, and the analysis of the data. The survey instrument, the aggregated data, and the prestudy (qualitative analysis of REF case studies) can be found on the project's website (https://www.impactdistillery.com/2020-impact-survey/). The survey was part of a BMBF-funded research project addressing the question of societal impact and its measurability in the social sciences and humanities (https://www.wihoforschung.de/de/impaqt-2631.php).

### Instrument

We designed a standardized instrument, consisting of six sections (A: sociodemographics, B: work context, C: knowledge transfer, D: teaching, E: research, F: general questions, G: personality). Here, we report on the largest section (C: knowledge transfer), which covers our three RQs, and use questions from sections A, B, and F for the explanatory analysis.

RQ1 was covered by a set of items (C1) addressing researchers' attitudes towards societal impact—i.e., their opinions regarding the importance of knowledge transfer as part of scientific activity and at their institutions, their opinions towards a stronger weighting of societal impact in research evaluation, and the performance of their institution's communication department in reaching relevant societal stakeholders. We used a 5-point Likert scale (from strongly disagree to strongly agree) to elicit agreement or disagreement with a statement.

RQ2 (goals) and RQ3 (formats) were implemented as multiple-choice questions (C3 and C5). The answer categories were based on a structuring content analysis of the REF impact case studies, which we carried out as a prestudy in spring 2019. In 2014, the UK became the first country to assess the societal impact of research as part of a national assessment. The REF evaluates societal impact via case studies, which are narratives that describe how research conducted at a higher education institution created a wider societal benefit. Of course, REF impact case studies are written to succeed in the evaluation in question [81]. Nevertheless, as researchers' expressions of impact, they provide a suitable textual data basis for developing goal and format categories [82, 83]. The REF case studies are available in a public database (https://impact.ref.ac.uk/casestudies/). The case studies were analyzed by two coders separately and frequently discussed in the research group in order to achieve intercoder reliability. The main categories of the content analysis were reflected in the questions on societal goals (i.e., What societal impacts do you want to achieve with your research?; 13 items—RQ2) and communication formats (i.e., Which transfer formats have you already used to communicate your results?; 6 items—RQ3). Each question allowed further responses in an open text field—from the few additional responses and the high response rate, we concluded that the identified categories were robust.

To ask about sociodemographic factors (sections A & B), we re-used questions that had already proved useful in a previous survey [77, 80]. The range of disciplines we asked about (A5) was based on the classification of the German Research Foundation [84].

### Pretests

To evaluate the quality and reliability of the survey, we conducted two rounds of pretests. The first round involved topic experts (i.e. fellow meta-researchers) and methodological experts who reviewed both the content and the design of the survey. The second pretest round

involved researchers from different disciplines who completed the survey, focusing on its usability and providing us with a first dataset for preliminary analysis and optimization. As a result, we revised descriptions and shortened the survey.

## Sample design and distribution

We designed a semi-convenience sample, which means that in principle any researcher with the link to the survey could participate. However, we personally invited certain researchers so that we could adequately cover the explanatory dimensions (i.e., the different organizational settings, disciplines, and career stages in Germany). With this in mind, we applied the following distribution strategy: We contacted the faculty heads of 60 German universities and 60 universities of applied science (*Fachhochschulen*; in the following, we use the short form "applied universities") and asked them to distribute the survey to researchers in their faculties. We selected the universities and applied universities based on the number of students and chose the 20 largest and the 20 smallest as well as 20 medium-sized ones. Additionally, we contacted the directors of each institute within the biggest German non-university research organizations (in the following shortened to "independent institutes"), i.e., the Max Planck Society, the Leibniz Association, the Helmholtz Association, and the Fraunhofer Gesellschaft. We also contacted the German Research Association's (DFG) graduate schools. Despite these efforts, our sample was not probabilistic and we assumed a certain self-selection bias due to the topic of the survey. Second, because the sample consisted of researchers in Germany, the transferability of the results to other research and innovation systems is limited.

The distinction between different types of research organizations (i.e., universities, applied universities, independent institutes) is important for this study. Independent institutes are characteristic of the German research system. They are typically independent of universities and focus on specific fields of research; scientists at these institutions are not obliged to teach. These institutes are typically organized within the Max Planck Society, the Helmholtz Association of German Research Centres, the Fraunhofer Society, and the Leibniz Association. Researchers from independent institutes should in theory be able to devote more resources to transfer activities than researchers at universities and applied universities, as this latter group has teaching obligations. Applied universities are Germany-specific tertiary education and research institutions. They are rather transfer-oriented and usually specialized in certain fields (e.g., arts, technology, or business). Researchers from applied universities typically have the most extensive teaching obligations. However, due to their applied approach, societal impact should presumably play a more central role for researchers at these kinds of institutions.

We conducted the online survey from April to June 2020. Participants were invited with an initial email and one reminder, both including a request to distribute the survey among colleagues.

## Sample description

Overall, 841 people started the survey, 534 of whom completed it (63%). In this paper, we focus on those who stated that their primary work location is Germany, leaving us with 499 valid (59%) cases to analyze.

Fig 1 provides an overview of the sample. Our distribution system had the desired effect—respondents from all disciplines and institutions took part. Regarding sociodemographic characteristics, the genders were equally distributed (50% male, 49% female, and 1% others). For ages 18 to 29 there were almost twice as many female as male researchers in the sample, while there were almost twice as many males as females in the 50 to 59 age group. For the 60+ group, there were approximately three times more male than female researchers in the sample (we

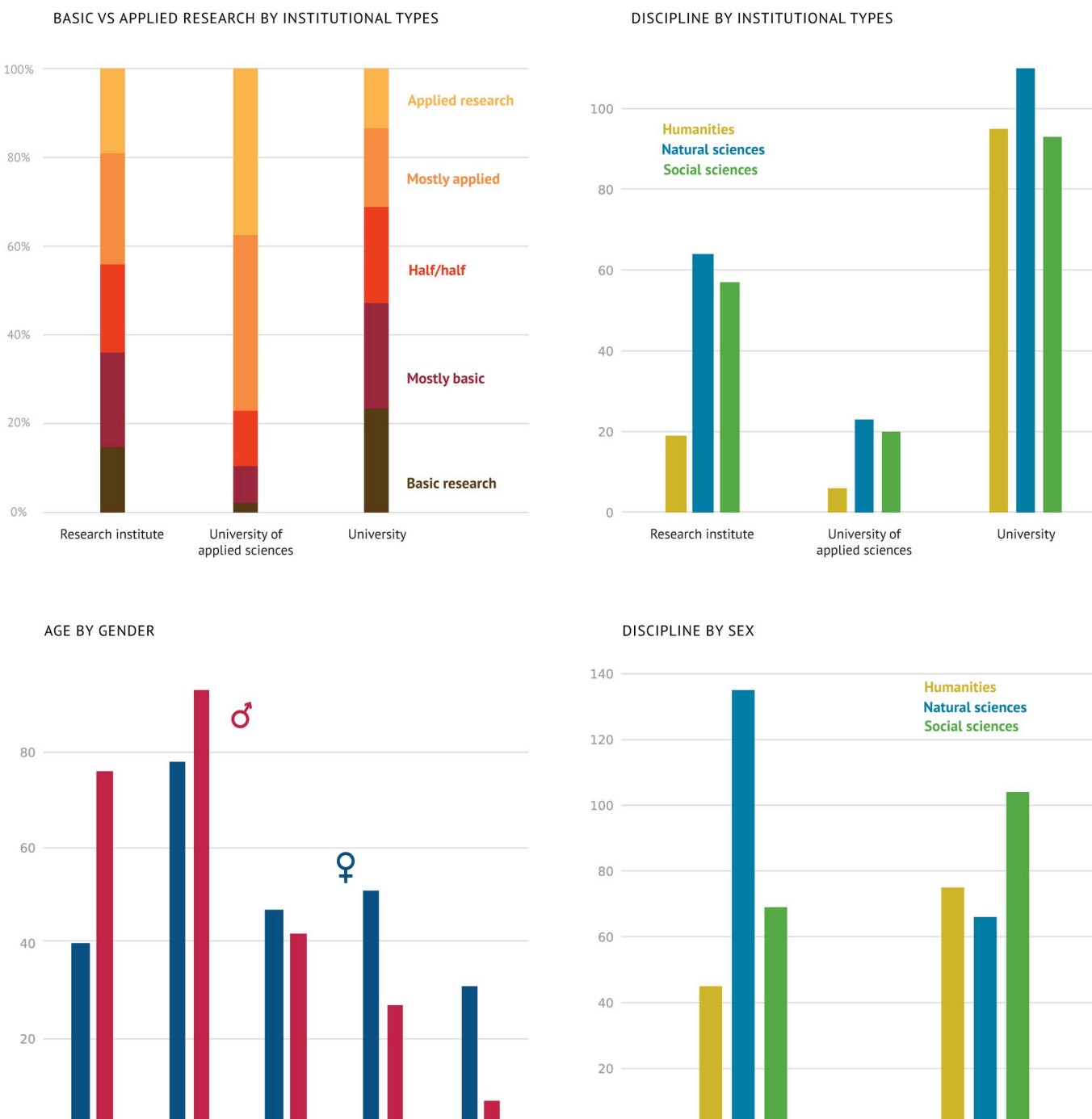

**Fig 1. Description of the sample using the explanatory variables that are at the core of our analysis.** (1) To what extent do researchers from the three disciplinary groups (natural sciences, social sciences, and humanities) consider their work as basic or applied research? (2) Distribution of the disciplinary groups across the prevailing institutional types. (3) Distribution of the researchers' sex across age groups. (4) Distribution of disciplinary groups for male and female researchers. The number of participants with other sex was so small that they were not included in these graphs.

excluded others due to the small sample size of 1%). This, however, corresponded roughly to the general gender distribution in German academia [85, 86].

60% of our participants worked at universities, 28% at independent institutes, 10% at applied universities, and only 2% at other forms of institutions; hence, when analyzing the organizational dimension, we focused on the three largest types of organizations. Furthermore, we found that 34% were PhD candidates, 25% were postdocs, 11% were academics without a PhD, 25% were professors and 5% were others. Regarding the disciplinary background, we used the classification of the German Research Association (DFG) and merged the disciplines into three groups: 41% of the respondents were from the natural sciences, 35% were from the social sciences, and 26% were from the humanities [84].

## Analysis

We found very little empirical research that looked at how researchers approach societal impact. Therefore, we conducted an exploratory analysis and focused on descriptive methods. Most of the results are presented as bivariate distributions that cross-tabulate the three dimensions from our conceptual framework with our research questions. For RQ1 (opinions on science communication), we asked respondents about evaluations because these would have practical implications for their working lives. With this particular dependent variable, we calculated a regression model (OLS) to analyze multivariate effects in more detail.

Some of the questions we analyzed were implemented as a 5-point Likert scale (strongly disagree, disagree, neutral, agree, strongly agree). In general, we considered agreement as the last two answer categories of the scale (agree + strongly agree) and report on this.

## Results

In the research section we present the results for the three research questions alongside the three dimensions of the independent variables, i.e., the content dimension, the organizational dimension, and the dimension of a researcher's individual characteristics.

### RQ1—opinion: Individual commitment without an institutional mandate

Overall, 89% of the respondents agree that public engagement is part of scientific activity. 53% agree that societal impact should be given more weight in research evaluations. Only 27% of respondents think that knowledge transfer plays an important role at their institution and that their communication department is reaching relevant stakeholders in society.

Across all disciplinary groups, the respondents largely agree that public engagement is part of scientific activity (from 86% in the natural sciences, 91% in the humanities, to 93% in the social sciences). Respondents vary in their opinions on whether societal impact should be given more weight in research evaluations: Only 40% of the respondents from the natural sciences agree compared to 65% for the social sciences and 58% for the humanities. Applied researchers are more in favor of societal relevance being given greater consideration in evaluations than basic researchers (see Fig 2). Only 15% of respondents from the humanities are convinced that their institutional communication departments are reaching relevant stakeholders in society, compared to 30% of natural scientists and 31% of social scientists.

Researchers at applied universities agree that societal relevance should have more weight in evaluation more often than those at independent institutes and universities—the approval rates are 62%, 49%, and 53% respectively. Researchers at universities show the lowest agreement to the statement that knowledge transfer plays an important role at their institution: Only 19% approve compared to 36% at applied universities and 37% at independent institutes. Furthermore, researchers at universities particularly disagree with the statement that their

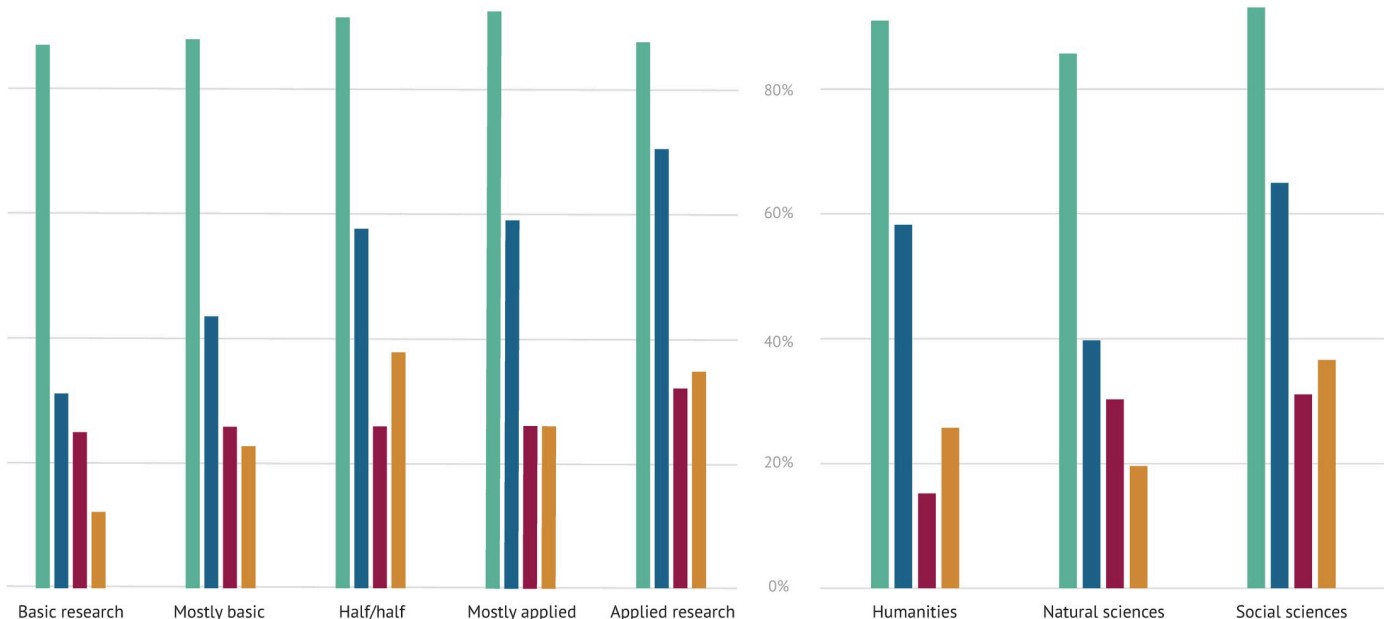

**Fig 2. Researchers' opinions on four core statements regarding societal impact.** The responses are differentiated by how much the researchers consider their work to be basic or applied research and the three disciplinary groups. The original question text was "How strongly do the following statements apply to you?." We combined the approving answer categories to calculate the approval rates.

communication departments are able to reach relevant stakeholders in society: Only 15% approve compared to 28% at applied universities and 44% at independent research institutes.

Both male and female researchers regard public engagement as part of scientific activity (89% and 90%). However, female researchers agree that societal relevance should be part of research evaluation more often than male researchers (62% compared to 44%). There are noteworthy differences among academic status groups: 60% of doctoral researchers but only 42% of postdocs and 47% of professors in our sample agree that societal relevance should have more weight in evaluation. Age does not seem to affect opinions on whether public engagement is part of scientific work.

Because of the differences outlined above, we decided to conduct a regression analysis (see Table 1). This confirms the descriptive observations: Humanities scholars (p = 0.01) and social scientists (p<0.01) are significantly more likely to agree that societal relevance should be part of research evaluation compared to natural scientists. Furthermore, female researchers (controlled for discipline), applied researchers, and younger researchers are significantly more in favor of including societal relevance in research evaluation.

This shows that researchers are generally in favor of societal impact and regard public engagement as part of scientific activity. Fewer researchers, but still many, agree that societal impact should have more weight in research evaluations. However, this assessment differs depending on the researcher's discipline, their applied or basic research focus, their gender,

**Table 1. OLS Regression on the researchers opinion regarding evaluation.**

| Dep. Variable: | Evaluation | R-squared: | 0.185 |
|---|---|---|---|
| Model: | OLS | Adj. R-squared: | 0.175 |
| Method: | Least Squares | F-statistic: | 18.39 |
| No. Observations: | 494 | Prob (F-statistic): | 2.68e-19 |
| Df Residuals: | 487 | Log-Likelihood: | -760.08 |
| Df Model: | 6 | AIC: | 1534. |
| Covariance Type: | nonrobust | BIC: | 1564. |

|  | coef | std err | t | P>|t| | [0.025 | 0.975] |
|---|---|---|---|---|---|---|
| Intercept | 0.5932 | 0.234 | 2.540 | 0.011 | 0.134 | 1.052 |
| Discipline: Humanities | 0.3468 | 0.136 | 2.546 | 0.011 | 0.079 | 0.614 |
| Discipline: Social Sciences | 0.4576 | 0.124 | 3.694 | 0.000 | 0.214 | 0.701 |
| Gender: Female | 0.3362 | 0.111 | 3.040 | 0.002 | 0.119 | 0.554 |
| Age | -0.1035 | 0.043 | -2.393 | 0.017 | -0.189 | -0.019 |
| Perceiving research as applied | 0.1760 | 0.036 | 4.902 | 0.000 | 0.105 | 0.247 |
| Considering science communication an academic task | 0.3539 | 0.062 | 5.724 | 0.000 | 0.232 | 0.475 |

How does the discipline, gender, age, a, applied-research focus, and the opinion that science communication is an academic task influence a researcher's opinion on whether or not societal relevance should be taken into account when evaluating research. The regression model was calculated using ordinary least squares. The reference category for the discipline is the natural sciences.

and their age. It is noteworthy that researchers (especially at universities) do not think that knowledge transfer plays an important role at their institution and that institutional communication departments are reaching relevant societal stakeholders. From this, we infer that societal impact is understood as an individual task for which there is no institutional mandate.

## RQ2—goals: Disciplines define societal goals

Overall, researchers' most important societal goal is to contribute to education (69%), followed by stimulating public discourse (55%). Equal proportions of respondents—37% each—regard contributing to informed political decision-making and to the physical and mental wellbeing of the population as a societal goal associated with their work. Given the choice of the 13 societal goals identified in the qualitative coding exercise, less than 10% of respondents selected contributing to national and/or international security and creating an entertainment offering; we hence excluded these in reporting. Fig 3 provides an overview of the societal goals the respondents could choose from by disciplinary group.

It is striking that the disciplinary groups have quite different impact profiles: Scholars from the humanities tend to have culture- and discourse-oriented goals, social scientists have discourse-, social-justice-, and policy-oriented goals, and natural scientists have technology-, health-, and environment-oriented goals. Making a contribution to education is by far the most common impact goal across all disciplines (87% for the humanities, 70% for the social sciences, 58% for the natural sciences). For humanities scholars and social scientists, stimulating and supporting public discourse is among the top societal goals (80% for humanities scholars and 74% for social scientists). Furthermore, 62% of humanities scholars consider preserving cultural heritage a goal of their activities; 54% of social scientists consider strengthening the position of disadvantaged groups to be an impact goal. The main goal for natural scientists is to drive technical innovation (59%), followed by making a contribution to education (58%), and contributing to the physical and mental wellbeing of the population (46%). The differences between applied and basic researchers in terms of their societal goals are negligible.

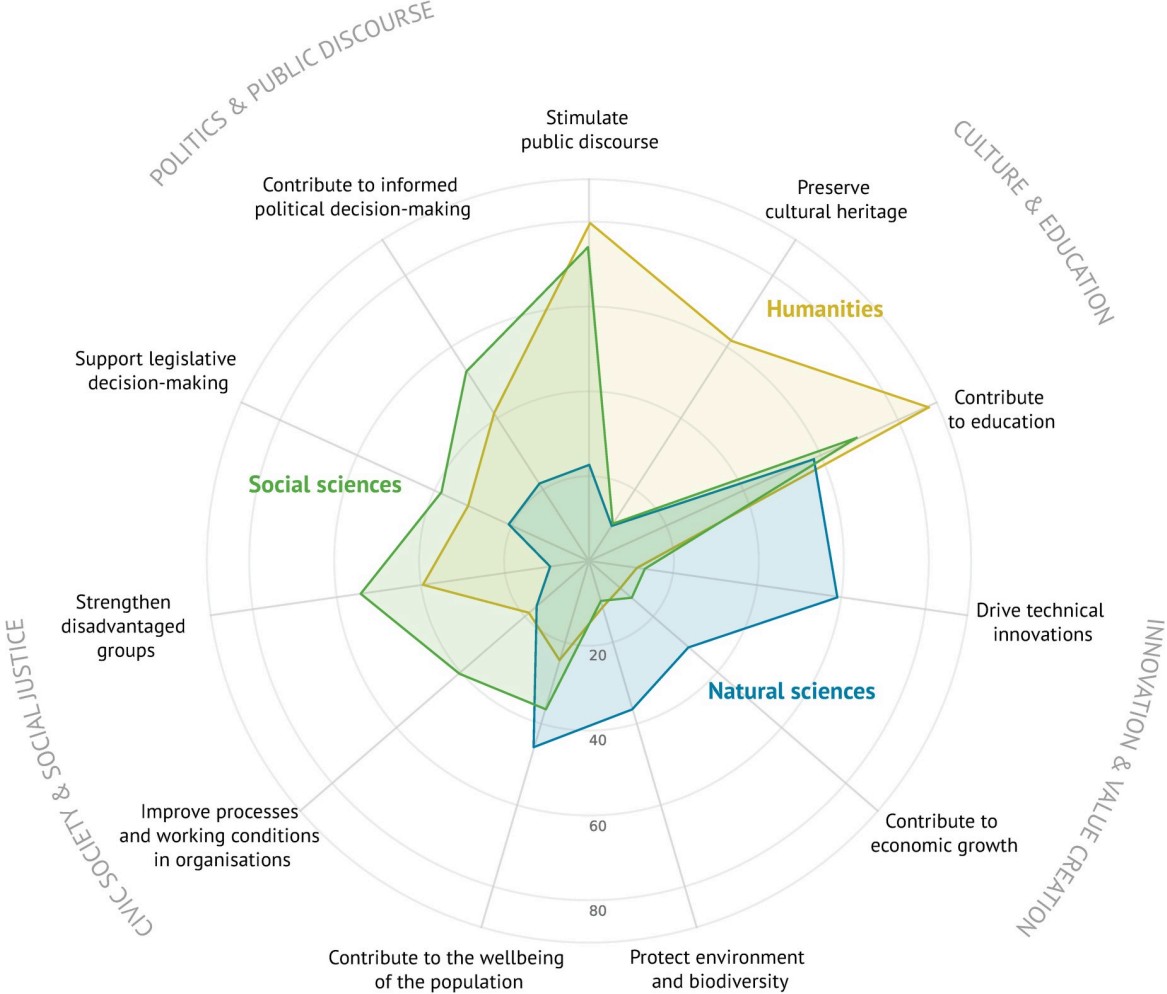

**Fig 3. Research goals by disciplinary groups.** The radar chart illustrates which goals researchers from our three disciplinary groups pursue when communicating with nonscientific audiences. The numbers are based on the question "What social effects do you most likely want to achieve with your research? With my research, I would like to . . .". The original question contained two more options (". . . contribute to national and/or international security." and ". . . create an entertainment offering."), but these were chosen by less than 10% of the respondents and are therefore excluded.

When looking at the different types of organizations, it is noticeable that researchers from applied universities are most economically oriented: 41% of the researchers from universities of applied sciences indicate that they want to contribute to economic value creation (19% at nonuniversity institutions and 17% at universities). In contrast, researchers from independent institutes are most the policy-oriented ones: 54% of the researchers from independent institutes aim to contribute to political decision-making (31% at universities and 37% at applied universities).

Young researchers (age group 18–29 years, mostly PhD students) reported less often that they aim to contribute to public discourse (37% in comparison to 65% for 30–39-year-olds) or

political decision-making (21% in comparison to 48% for 40–49-year-olds). When it comes to gender, male researchers are more inclined to pursue goals that are also related to their disciplines and vice versa. For example, technical innovation is a goal for 43% of male researchers and for 20% of female researchers. Female researchers are more interested in supporting minorities (41%) than male researchers (25%). The academic status of the respondents did not have any noteworthy effects on their goals.

From this, we conclude that societal goals are best explained by the disciplinary backgrounds of researchers. While natural scientists chose goals in the spheres of environment and health as well as innovation and economic value creation, social scientists' goals centred more on civic society and social justice as well as politics and public discourse. Humanities scholars also chose discourse-oriented goals and focused on societal goals in the cultural sphere. Despite the differences in the definition of societal goals, it is noteworthy that making a contribution to education ranks among the top three goals for every disciplinary group. It is also evident that researchers from applied universities tend to be more economically oriented, while researchers from independent institutes tend to be more policy-oriented.

### RQ3—formats: University researchers are the least active

The only transfer format that more than half of the respondents (68%) had used in the past are events (i.e., including public lectures, exhibitions, expert panel discussions). The second most used communication format is public relations (i.e., comment pieces in newspapers or interviews)—45% of respondents have used these in the past. This is followed by educational offerings in schools and for civil society (43%), social media communication (38%), advisory formats (33%), and collaborations with nonscientific partners (33%).

It is evident that humanities scholars especially use social media to communicate about their research: 55% of humanities scholars approve of such methods compared to 33% of natural scientists and 38% of social scientists. Social scientists have the most experience with advisory formats: 50% have used them compared to 28% of humanities scholars and 24% of natural scientists. Comparing applied and basic researchers, we noted that basic researchers hardly ever offer advisory formats (14% for basic researchers vs. 47% for applied researchers) or collaborations (17% for primarily basic researchers vs. 39% for primarily applied researchers).

Researchers from applied universities are the primary users of advisory formats: 57% of researchers at applied universities have used advisory formats compared to 40% of researchers at independent institutes and only 26% of the researchers at universities. Only 28% of the researchers from universities have used collaboration formats, compared to 53% of researchers from applied universities and 35% of researchers from independent institutes. As Table 2 shows, university researchers have remarkably low scores on every communication format.

**Table 2. Based on the question "which transfer formats have you already used to communicate your results?" The table indicates which transfer formats researchers from the three prevailing institutional types use.**

| | University | Applied university | Independent institute |
|---|---|---|---|
| **Education offerings** (e.g., for schools, civil society groups) | 43.6% | 59.2% | 39.3% |
| **Advisory formats** (e.g., reports for politicians/public administration/companies/NGOs) | 25.8% | 57.1% | 40.7% |
| **Events** (e.g., public lectures, exhibitions, expert panel discussions) | 61.4% | 71.4% | 79.3% |
| **Public relations** (e.g., through comments in newspapers, interviews, appearances in TV programs) | 38.6% | 59.2% | 55.0% |
| **Social media communication** (e.g., podcasts, Twitter) | 31.9% | 22.5% | 54.3% |
| **Collaborations with nonscientific partners** (e.g., citizen science, industry partnership) | 27.9% | 53.1% | 35.0% |

Concerning researchers' individual characteristics, the analysis shows that social media platforms are used more by younger scientists. Less surprisingly, the older a researcher is, the more likely it is that he or she has ever used a format. Regarding gender differences, more male than female researchers have used advisory formats: 43% of male researchers have done so compared to 24% of female researchers.

To conclude, we find that researchers use a variety of formats to communicate results. There are, however, some interesting differences in usage: Humanities scholars particularly tend to communicate their research via social media. Social scientists are most experienced when it comes to advisory formats. Researchers from applied universities are especially likely to have used advisory and collaboration formats. Younger researchers use social media to communicate about their research more than older researchers, and male researchers particularly tend to use advisory formats. In general, it is noteworthy that university researchers are the least active group for almost any of the formats.

## Discussion and conclusions

Societal impact, and hence what academic research offers the economy, society, culture, public administration, health, environment and overall quality of life, is gaining in importance in science governance [2, 5, 6]. Likewise, the topic is on the top of the agenda of prominent policy makers and research organizations in Germany, where our study took place. If science is to be assessed based on its contribution to society, the conditions under which social impact arises should be clear. This article contributes to this discourse by addressing researchers' perspectives on societal impact, that is, their opinions on societal impact (RQ1), the societal goals they associate with their research (RQ2), and the formats they use to engage with society (RQ3). For this reason, we conducted a survey among 499 researchers in Germany from April to June 2020.

Regarding researchers' opinions, it is remarkable that the majority of researchers (89%) consider societal engagement to be part of scientific activity. More than half of the researchers (53%) agree that societal impact should be given more weight in evaluations. Even though the majority of researchers regard public engagement as part of scientific work, they are not equally positive about whether societal impact should have more weight in evaluations. One reason for this discrepancy may be that researchers fear that evaluations will lead to additional work or that they will not adequately record their transfer activities [23, 60, 64]. In addition, it is striking that only 27% of the respondents assume that knowledge transfer plays an important role at their institution; also 27% n believe that the institutional communication department is managing to reach relevant stakeholders in society. Humanities scholars (15%) and university researchers (15%) particularly doubt that their communication departments are reaching relevant societal stakeholders. This mirrors previous findings suggesting a certain decoupling between central transfer infrastructures and researchers [38, 39] and leads us to hypothesize that there is a certain mismatch between individual and institutional commitment.

Regarding the societal goals that researchers associate with their work, it is noteworthy that contributing to education is by far the most important goal (picked by 69% of the respondents) across all groups, followed by stimulating public discourse (55%) and contributing to informed political decision-making (37%). Moreover, it is apparent that societal goals are subject to disciplinary considerations: We show that scholars from the humanities have culture and discourse-oriented goals, social scientists have discourse-, policy- and social-justice-related goals and natural scientists have innovation- and health-related goals. This supports many theoretical reflections about the epistemic conditions for societal impact and the different roles that disciplines occupy in society [36, 56, 87]. We also find that young researchers (age 18–29) are

less keen on stimulating public discourse than older ones (37% in comparison to 65% for 30–39-year-old researchers) or informing political decision-making (21% for young researchers compared to 48% for 40–49-year-old researchers). This can likely be explained by the experience that older researchers have acquired, which might make them especially well placed to influence public and political decision-making.

As far as the formats used for societal outreach are concerned, the most commonly used ones are events (used by 65% of the respondents), followed by public relations via media (45%), and educational formats for schools and civil society groups (43%). Humanities scholars especially use social media to communicate results (55% compared to 33% of natural scientists and 38% of social scientists). Basic researchers do not use advisory formats (14% vs. 47%) or collaborations (17% vs 39%) anything as often as applied researchers. Note that university researchers report remarkably low usage of any communication format compared to researchers at applied universities or at independent institutes. This might be explained by the fact that university researchers have teaching obligations not faced by those at independent institutes and thus less time for engagement activities. However, researchers from applied universities typically have a higher teaching workload. As far as researchers' individual characteristics are concerned, it is notable that social media is used more by younger researchers, indicating that social media will likely become more important as a means of engagement.

While our results are mainly of a descriptive nature and we do not make normative assumptions about the subject of our research (i.e. "it is good to have more impact"), we can still draw some practical conclusions: First, considering the discontent with institutional communication departments, it might be worthwhile to implement decentralized support structures on the mesolevel of research organizations. This could more adequately address the complexities of the sciences and their many publics [5, 6, 57, 63]. The findings further suggest that, where applicable, organizational factors (e.g., institutional investments in transfer, training offerings, support infrastructures) should be more strongly incorporated into assessments of societal impact—for example. through formative evaluations [88]. Second, our results suggest that it is strongly advisable that evaluation exercises are responsive to disciplinary differences. For example, if economic and technical impact were the sole basis for assessing societal impact, social sciences and humanities scholars would be discriminated against [6, 63]. Our framework for societal goals and our results can also be the basis for disciplinary self-understanding (e.g., in learned societies), in that they can stimulate a normative discussion about good transfer and its evaluation. Third, considering the comparatively low importance of social media as a means of communicating about research, care should be taken not to overuse online discourse as a way of easily generating impact proxies. Moreover, our findings contribute to an informed discussion in science governance about the constraints of impact evaluation and might help impact officers and communication professionals at universities to reflect on strategies.

Our results add to the ongoing scholarly discourse on societal impact. We think that our results could bring two relevant but still separated discourses closer together, that is, the discourse on impact evaluation and the discourse on science communication/public engagement. It makes sense for critical evaluation research to make use of empirical work on the exchange between science and society. Vice versa, it makes sense to examine how evaluation policies for societal impact might affect researchers' communication behavior. We further provide initial evidence on potentially relevant research perspectives. This concerns the organization of societal impact at scientific institutions. In addition, we suggest that national innovation systems should be studied comparatively in order to understand the impact of policies and to study more closely the relationship between transfer practices and societal impact in specific disciplines.

## Acknowledgments

We would like to thank Natalya Sokolovska, Sascha Schönig, and Elias Koch for their help in coding the REF impact case studies. We would also like to thank Gert G. Wagner and Ricarda Ziegler for their friendly feedback.

## Author Contributions

**Conceptualization:** Benedikt Fecher, Marcel Hebing.

**Data curation:** Benedikt Fecher, Marcel Hebing.

**Formal analysis:** Benedikt Fecher, Marcel Hebing.

**Funding acquisition:** Benedikt Fecher, Marcel Hebing.

**Investigation:** Benedikt Fecher, Marcel Hebing.

**Methodology:** Benedikt Fecher, Marcel Hebing.

**Project administration:** Benedikt Fecher, Marcel Hebing.

**Resources:** Benedikt Fecher, Marcel Hebing.

**Software:** Benedikt Fecher, Marcel Hebing.

**Supervision:** Benedikt Fecher, Marcel Hebing.

**Validation:** Benedikt Fecher, Marcel Hebing.

**Visualization:** Benedikt Fecher, Marcel Hebing.

**Writing – original draft:** Benedikt Fecher, Marcel Hebing.

**Writing – review & editing:** Benedikt Fecher, Marcel Hebing.

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
