## [Decision Letter · Decision Letter 0]

24 Mar 2021

PONE-D-21-03474

How Do Researchers Achieve Societal Impact? Results of an Empirical Survey Among Researchers in Germany

PLOS ONE

Dear Dr. Fecher,

Thank you for submitting your manuscript to PLOS ONE. After careful consideration, we feel that it has merit but does not fully meet PLOS ONE’s publication criteria as it currently stands. Therefore, we invite you to submit a revised version of the manuscript that addresses the points raised during the review process.

Reviewers agreed that the paper is valuable; however, they suggest several revision points. I advise you to incorporate as many suggestions as possible, and when not possible, please explain why. Please enhance the method section by clearly explain the statistics and the survey. A more detailed introduction and discussion are needed.

We look forward to receiving your revised manuscript.

Kind regards,

Laurentiu Rozylowicz, Ph.D.

Academic Editor

PLOS ONE

Journal Requirements:

2/ Please improve statistical reporting and refer to p-values as "p<.001" instead of "p=.000". Our statistical reporting guidelines are available at https://journals.plos.org/plosone/s/submission-guidelines#loc-statistical-reporting

3. Please include a copy of Table 3 which you refer to in your text on page 19.

4. Please ensure that you refer to Figure xxxxx in your text as, if accepted, production will need this reference to link the reader to the figure.

Reviewers' comments:

Reviewer's Responses to Questions

**Comments to the Author**

1. Is the manuscript technically sound, and do the data support the conclusions?

Reviewer #1: Partly

Reviewer #2: Yes

2. Has the statistical analysis been performed appropriately and rigorously? 

Reviewer #1: No

Reviewer #2: Yes

3. Have the authors made all data underlying the findings in their manuscript fully available?

Reviewer #1: Yes

Reviewer #2: Yes

4. Is the manuscript presented in an intelligible fashion and written in standard English?

Reviewer #1: No

Reviewer #2: Yes

5. Review Comments to the Author

Reviewer #1: The research topic of the study is relevant both academically and socially. The authors bring out the important question how researchers understand their efforts to engage in non-academic organizations and institutions and their possibilities to achieve social impact via their research organizations. They point out rather interesting results. Statistical research on research impact is clearly underdeveloped and needs more studies like this. However, the manuscript has several major problems in methodical soundness and in the theoretical framework, which is not connecting to the results or to the academic discussion about research impact.

There are four major problems that the paper needs to address:

1. The whole analysis section is missing regarding the statistical analysis. The reader cannot see how the authors conducted the study. There is only one section where the authors uses a regression analysis, while the whole analysis is descriptive. It needs a good reason to ground such an analytical choice. The statistical description requires re-phrasing and the tables need reformulations. It is rather difficult to understand the authors’ writing through these expressions and the table.

2. Because the questionnaire of this research is self-designed, there should be description of its validity and reliability, which are now missing.

3. Because the method, analysis and results are insufficiently presented, I strongly recommend to ask an expert in statistics to re-check and revise these sections. I believe such expertise would help to improve the readability, validity and reliability of this article.

4. The theoretical framework relating to science communication in the manuscript does not connect to the results of the study. The discussion section lacks correspondence to academic discussion about research impact. There are several theoretical approaches to social impact of research or science communication and the authors should point out what point of view their hypothesis and results are supporting. The interpretation of the results lacks a meaningful theoretical framework, although the study presents a framework for science communication. I recommend to structure a hypothesis vis-à-vis science communication and interpret the results by this framework for the study.

I suggest some minor recommendations here:

- I recommend that a professional language service would proof read this article.

- The title of the paper corresponds weakly to the study, as the paper studies people’s opinions and understanding of their engagement, not how they actually achieve impact.

- The study utilizes the categories of the Research Excellence Framework (REF) from the United Kingdom. However, the REF cannot be argued to be without normative assumptions when it is a result of policies. The utilization in analysis needs some theoretical justification for using such criteria for impact.

- Basic and applied distinction is not always clear and divergent disciplines may have differing opinions about this. Use of such distinction needs theoretical support.

- The questionnaire has a section considering respondent’s personality. I cannot see this data in the paper and the meaning of such data for the paper.

- What do the authors mean by: "...by the socio-demographic factors: gender, status, and age."? In addition what “status”, do you mean employment status? Please, be more specific of these terms.

- Expression “In relation to the engagement with society” is unclear. It can be read that research is not part of society.

- Did you select the universities according to your criteria for the selection among all universities in Germany?

- Your description of semi-convenient/convenient sample is not consistent in the paper.

- The result section is rather long, and some parts are quite repetitive. Shortening it could make it more reader-friendly.

- Description of the REF coding needs to be placed in the method section.

- Table 2 title needs to be more specific.

- Table 3 is missing completely.

- What does “sustainable strategy for impact” mean in this context?

- What is specific impact in relation to disciplines? I don't think reference to a policy paper is relevant in discussion part. It should have discussion with the science communication literature or impact literature.

- In the appendix figure, there is a wrong title for the bar chart: basic vs. applied.

Reviewer #2: Thanks for the opportunity to review your manuscript entitled "How do researchers achieve societal impact? Results of an empirical survey among researchers in Germany" in which you have identified and argued for an important research gap. I really enjoyed reading the manuscript and my main comments and suggestions below are merely about elaborating and developing your text a bit more.

Consider the following in your revision of the paper:

INTRODUCTION

In the section SOCIETAL IMPACT IN RESEARCH p 10 ff you may problematize and position your study

more by (i) a deeper description existing research of the different concepts of academic knowledge

creation see e.g., Olsson et al 2020, (2020). A conceptual model for university-society

research collaboration facilitating societal impact for local innovation. European Journal of Innovation

Management. https://doi-org.ezproxy.server.hv.se/10.1108/EJIM-04-2020-0159 and (ii) earlier

research on measuring and defining research impact e.g., De Jong, et al (2014) “Understanding

societal impact through productive interactions: CT research as a case”, Research Evaluation, Vol. 23

No. 2, pp. 89-102; Greenhalgh, T., Jackson, C. et al (2016), “Achieving research impact through co-

creationin community-based health services: literature review and case study”, The Milbank

Quarterly, Vol. 94 No. 2, pp. 392-429; Matsumoto, M., et al (2010), “Development of a model to estimate the economic impacts of R&D output of public research institutes”, R & D Management, Vol. 40 No. 1, pp. 91-100. Issues of co-creation in university-society collaboration is also something you may find interesting see e.g., Larsson, J. and Holmberg, J. (2018), “Learning while creating value for sustainability transitions: the case of challenge lab at Chalmers university of technology”, Journal of Cleaner Production, Vol. 172, pp. 4411-4420, and Olsson et al 2020 as above.

AIM – there is a slight difference between the aim on page 5 and page 19.

RQ1 includes the words engagement, evaluation and press departments that need further explanation and linked to earlier research in the introduction section.

This is also reoccurring in the Method section p. 7 as the importance of communication and

performance of communication department- is there any earlier research on these aspects then it should be mentioned in the Introduction.

P. 6 define status in the individual dimension.

P. 12 The status groups -presented differently throughout the paper. A logical presentation of the groups is recommended along with an early definition in the paper of how you define a ‘researcher’.

The age of 18 seem to be very young to be included in the group of researchers?

Methodology and results are presented and illustrated in a proper way.

Fig 1 Institutional types- may the categories by merged into three categories instead?

The manuscript ends with a Discussion – add Conclusion in the heading or add another subsection entitled Conclusion. There are very few references in your Discussion. Return to your earlier refences in the Introductions section and compare your findings to earlier research.

Good luck with your revision!

6. PLOS authors have the option to publish the peer review history of their article (what does this mean?). If published, this will include your full peer review and any attached files.

Reviewer #1: **Yes: **Juha-Pekka Lauronen

Reviewer #2: No

---

## [Author Response · Author response to Decision Letter 0]

8 May 2021

Dear Dr. Rozylowicz, dear colleagues,

we were delighted about the positive and helpful reviews we received. We substantially revised the manuscript accordingly.

Sincerely,

Benedikt Fecher

---

## [Decision Letter · Decision Letter 1]

28 May 2021

PONE-D-21-03474R1

How Do Researchers Approach Societal Impact?

PLOS ONE

Dear Dr. Fecher,

Thank you for submitting your manuscript to PLOS ONE. After careful consideration, we feel that it has merit but does not fully meet PLOS ONE’s publication criteria as it currently stands. Therefore, we invite you to submit a revised version of the manuscript that addresses the points raised during the review process.

The revision greatly improved the paper, and both reviewers had positive comments. However, there is a need for some minor adjustments for greater clarity and better reproducibility. Please try to include the suggestions in your paper, and, if not suitable, please answer in detail why. Also, please proofread the paper carefully before submitting it.

We look forward to receiving your revised manuscript.

Kind regards,

Laurentiu Rozylowicz, Ph.D.

Academic Editor

PLOS ONE

Journal Requirements:

Reviewers' comments:

Reviewer's Responses to Questions

**Comments to the Author**

1. If the authors have adequately addressed your comments raised in a previous round of review and you feel that this manuscript is now acceptable for publication, you may indicate that here to bypass the “Comments to the Author” section, enter your conflict of interest statement in the “Confidential to Editor” section, and submit your "Accept" recommendation.

Reviewer #1: (No Response)

Reviewer #2: All comments have been addressed

2. Is the manuscript technically sound, and do the data support the conclusions?

Reviewer #1: Yes

Reviewer #2: Yes

3. Has the statistical analysis been performed appropriately and rigorously? 

Reviewer #1: Yes

Reviewer #2: Yes

4. Have the authors made all data underlying the findings in their manuscript fully available?

Reviewer #1: Yes

Reviewer #2: Yes

5. Is the manuscript presented in an intelligible fashion and written in standard English?

Reviewer #1: Yes

Reviewer #2: Yes

6. Review Comments to the Author

Reviewer #1: PONE-D-21-03474R1

How Do Researchers Approach Societal Impact?

PLOS ONE

Review

The

The paper and its methodical choices have improved significantly, and that is why, I suggest only some minor revisions by pointing out small possible improvements and recommendations. The study design still has some minor questions related to validity and reliability, which could be clarified and elaborated more. The findings and conclusions of the study are a bit trivial because they do not debate openly with the previous literature and the background literature of the study. Connection to the previous debate could elevate the substantial value of the paper.

1. Pretests: The authors explain that they have done pretests for reliability, but they do not elaborate what result the test provided for reliability, though they explain what they did based on those tests.

2. In the sample design, the authors describe that they selected the universities based on number of students. Why do they not rely it on researchers, teachers and other staff of scholars when the study is about researchers’ attitudes?

3. In the page 19, the expression sounds ambiguous: “Fewer researchers, but still many,…”

4. In the results section (RQ 1) the authors conclude that: “It is noteworthy that researchers (especially at universities) do not think that knowledge transfer plays an important role at their institution and that institutional communication departments are reaching relevant societal stakeholders. From this, we infer that societal impact is understood as an individual task for which there is no institutional mandate.

It is a bit unclear in this argument whether it means understanding of these institutions or the understanding of the researchers in the survey. If this means only the research staff’s attitudes, I suggest to formulate the conclusion differently, so that it is clear that it describes only the staff’s believes. Of course these two dimensions could be compared preferably.

This conclusion of neglected organizational aspects of impact requires information about the actual knowledge transfer activities of the survey institutions, and how much emphasis they have on impact evaluation and practices. Now the authors provide information about general policy recommendations and considerations, but this does not describe what kind of practices the universities have regarding impact requirements, their organization and evaluation, and their relationship with the survey attitudes.

Perhaps, the authors could provide some basic information of how impact is considered in the university and research work in Germany.

5. The claim “but these were chosen by less than 10% of the respondents

and are therefore excluded” is mentioned twice in the same context.

6. In the results (RQ2), the authors claim that “It is striking that the disciplinary groups have quite different impact profiles: Scholars from the humanities tend to have culture- and discourse-oriented goals, social scientists have discourse-, social-justice-, and policy-oriented goals, and natural scientists have technology-, health-, and environment-oriented goals.

This claim is one of the main results in the paper. However, there is nothing striking in this argument according to common knowledge of disciplinary practices and recent studies on research impact and public engagement. Perhaps, the authors could emphasize how these findings support previous understanding of the characteristics of these disciplines and their public engagement.

7. Then the authors conclude that: “From this, we conclude that societal goals are best explained by the disciplinary backgrounds of researchers.”

Because several authors have already pointed out this argument in recent literature (e.g. Benneworth) and in older classics (e.g. Weiss), the authors could elaborate this statement by linking it to recent literature, which is worried about impact policies neglecting such disciplinary goals and their importance.

8. The results (RQ3) bring out education offerings as an important perception of impact. I was wondering why this does not seem to consider the education of the institutions themselves, as they provide a great number of new graduates for employment each year.

9. Usually some sort of limits of the study should be brought out in the conclusions.

10. The conclusions seem a bit unfinished. Maybe the main conclusion is that disciplines have had their own impact goals in relation to contemporary society all along, and now all the responsibilities are suddenly pressured against researchers without proper institutional structures. On the other hand, what are the impact goals of the organizations and do they correspond to disciplines’ goals? As I pointed out in the sixth comment, linking the findings to recent studies on the disputes about impact goals and neglection of disciplinary goals, to older discussion about the difference between research utilization of various academic fields and to the structural/organizational levels of research impact could benefit and increase the value of the paper.

10. The authors point out: certain mismatch between individual and institutional commitment. However, what are the actual institutional demands from individuals?

11. I find this interesting: "Humanities scholars especially use social media to communicate results."

Perhaps, the authors could speculate a bit why the survey has such a surprising result.

12. This is unclear: researchers have teaching obligations not faced by those at independent institutes and thus less time for engagement activities. However, researchers from applied universities typically have a higher teaching workload.

What does this mean regarding engagement activities?

13. I find this statement unnecessary, as the paper clearly has a normative tone: While our results are mainly of a descriptive nature and we do not make normative assumptions about the subject of our research (i.e. “it is good to have more impact…

14. “The findings further suggest that, where applicable, organizational factors (e.g., institutional investments in transfer, training offerings, support infrastructures) should be more strongly incorporated into assessments of societal impact”

I don’t understand how the authors came to this conclusions. Why should they be part of assessments of impact? Why is impact assessment the perfect way to promote public engagement? There are also other ways to organize and promote knowledge in society.

15. This gives a misleading impression that innovation is same than impact: “In addition, we suggest that national innovation systems should be studied comparatively in order to understand the impact of policies and to study more closely the relationship between transfer practices and societal impact in specific disciplines.”

Some minor notions:

- “An example of this is Jong et al.'s concept of productive interactions…”

Productive interactions is a concept of Spaapen and van Drooge (2011).

- “According to Bornmann, three main strands of societal impact can be distinguished:”.

This is probably de Jong et al. (2014) interpretation based on Bornmann’s conceptualization.

- The instrument: “Nevertheless, as researchers’ expressions of impact, they provide a suitable textual data basis for developing goal and format categories”

Some studies (Watermeyer et al.) have shown that the REF case studies construct unsophisticated imaginaries of impact. This problematics could be mentioned regarding the evaluation competition.

Reviewer #2: (No Response)

7. PLOS authors have the option to publish the peer review history of their article (what does this mean?). If published, this will include your full peer review and any attached files.

Reviewer #1: **Yes: **Juha-Pekka Lauronen

Reviewer #2: No

---

## [Author Response · Author response to Decision Letter 1]

14 Jun 2021

Again, thank you for this valuable and constructive feedback which we feel added substantially to the quality of the paper!

---

## [Editor Report · Decision Letter 2]

18 Jun 2021

How Do Researchers Approach Societal Impact?

PONE-D-21-03474R2

Dear Dr. Fecher,

We’re pleased to inform you that your manuscript has been judged scientifically suitable for publication and will be formally accepted for publication once it meets all outstanding technical requirements.

Kind regards,

Laurentiu Rozylowicz, Ph.D.

Academic Editor

PLOS ONE
---

## [Editor Report · Acceptance letter]

1 Jul 2021

PONE-D-21-03474R2 

How do researchers approach societal impact? 

Dear Dr. Fecher:

I'm pleased to inform you that your manuscript has been deemed suitable for publication in PLOS ONE. Congratulations! Your manuscript is now with our production department. 

Kind regards, 

on behalf of

Dr. Laurentiu Rozylowicz 

Academic Editor

PLOS ONE